# Diabetes Mellitus and Its Risk Factors among Migrant Workers in Kuwait

**DOI:** 10.3390/ijerph19073943

**Published:** 2022-03-25

**Authors:** Anwar Ali, Shaikhah Alfajjam, Janvier Gasana

**Affiliations:** 1Public Health Department, Ministry of Health, P.O. Box 5, Kuwait City 12009, Kuwait; anwar.althamer@gmail.com; 2Occupational Health Department, Ministry of Health, P.O. Box 5, Kuwait City 12009, Kuwait; shaikhah.alfajjam@hotmail.com; 3Department of Environmental and Occupational Health, Faculty of Public Health, Kuwait University, P.O. Box 24923, Kuwait City 13110, Kuwait

**Keywords:** diabetes, migrants, diabetes risk factors

## Abstract

The prevalence of diabetes mellitus (DM) is growing enormously worldwide, and actions need to be taken in order to minimize the burden of diabetes mellitus and reduce its complications. Since two-thirds of Kuwait’s population are expatriates, the prevalence of and factors associated with diabetes among migrant workers was assessed as it has a significant impact on migrant workers’ quality of life, health, and productivity. The data used in this study was for all migrant workers who attended Shuaiba Industrial Medical Center (SIMC) for physical examination in the year 2018. Univariate and multivariate regression were used to assess the relationship between diabetes mellitus and the other independent factors where odds ratios with confidence intervals were delineated. Information for a total of 3477 participants was recorded in the dataset for 2018. Of the total participants, 10.1% had diabetes mellitus. About 49% of the participants were overweight. The largest age group of participants was between 31 and 40 years of age. A small percentage of the participants were diagnosed with hypertension at 11.8%. Additionally, 76.1% of the participants reported themselves as non-smokers. Diabetes was positively associated with age, hypertension, and nationalities. However, no association was found between BMI and smoking tobacco. This is the first study in SIMC to assess DM and its associated risk factor among migrants, since migrant workers are neglected subpopulations that need our focus and attention to achieve justice and fairness. The findings revealed that the prevalence of DM among our study population was considerably lower. However, a healthy lifestyle, including a healthy diet and being physically active, need to be introduced to prevent any further damage.

## 1. Introduction

Diabetes mellitus (DM) is one of the most common preventable chronic diseases worldwide that is associated with abnormal levels of sugar in the blood. This may be due either to the inadequate insulin production in the body or to the cells that are no longer sensitive to perform their actions [1]. Diabetes can lead to higher mortality rates and many serious complications, such as kidney problems, cardiovascular diseases, blindness, and many other health conditions, that are expensive to manage and thus result in reducing the quality of life [2].

The prevalence of DM worldwide, according to the International Diabetes Federation (IDF), was around 425 million people aged between 20 and 79 years old in the year 2017, with an expectation to rise to 629 million people who will be living with diabetes by the year 2045 [3]. Locally, Kuwait is ranked as one of the top twenty countries worldwide that have the highest prevalence of diabetes among adults, after age adjustment, where about 22% of adults in Kuwait have diabetes [4].

Migrant workers reached more than 160 million people worldwide [5]. For example, In Kuwait, the number of migrant people is almost 3 million of the total population, which is 4.5 million [5]. As reported by Alahmad et al., unlike local citizens, migrant workers are considered unprotected subpopulations which are being neglected by public policies. Different stressors affect migrant workers’ lives: stressors from workplace, environment, community, and individual stressors [5]. As per Alahmad’s study [6], in Kuwait, migrant worker communities‘ differential access to education, housing, healthcare, and employment reduces their quality of overall health. Migrant workers who make up the majority of the population are employed in low-skilled labor jobs and domestic work, facing precarious working conditions, financial hardships, and inadequate housing. Having little knowledge of the health insurance systems and language barriers makes it difficult for them to access healthcare when needed. Additionally, migrants are more likely to work in occupations that increase the risk of transmission and are generally excluded from protections in public policies. All these factors contribute to a structural and systematic health disadvantage that gives rise to health disparities in Kuwait amongst ethnic populations. Furthermore, diabetes mellitus and its related chronic conditions may aggravate the health of migrant workers. According to the International Labor Organization (ILO), diabetes and its complications make up most of the worker absenteeism [7].

Correspondingly, the risk of developing diabetes depends on several modifiable and non-modifiable risk factors. Increasing age, some specific ethnicity, lower socioeconomic factors, male gender, history of smoking, history of poor control of blood glucose, pancreatic diseases, genetic predisposition, and a family history of diabetes are considered the main non-modifiable risk factors [8]. On the other hand, obesity, hypertension, physical inactivity, abnormal cholesterol level, and smoking are all modifiable risk factors [9].

The prevalence of diabetes type 2 is increasing enormously worldwide, and its impact can be seen as an epidemic on both human health and the health economy. Therefore, preventing diabetes should be a high public health priority, especially since diabetes type 2 is largely attributed to the wrong daily decisions that people make, as it is mainly associated with an unhealthy lifestyle. Thus, this study aims to assess the prevalence of diabetes type 2 and its associated risk factors among migrant workers in Kuwait, to draw recommendation for a better health outcome among this group of people and better prevention and control measures of such a disease.

## 2. Subjects and Methods

### 2.1. Study Population

The study is a retrospective cross-sectional study where data was obtained from medical records of the Shuaiba Industrial Medical Center (SIMC) at the department of Occupational Health of the Ministry of Health in Kuwait. SIMC, as an occupational health service provider, which was established in 1982, aims at protecting, promoting, and maintaining good health among workers. SIMC provides occupational medical preventive services, in addition to the primary medical care as well as emergency services for the workers from the industrial areas in their coverage, which include Al Shuaiba industrial area, Abdulla Port, and Al Ahmed area. SIMC offers different types of examination to workers by which data were collected, which are: pre-employment examination and periodic examination. Since the study focuses on migrant workers, SIMC was considered an ideal site to study the prevalence of diabetes among workers due to its unique services provided.

### 2.2. Data Collection

This secondary data in this study was obtained from SIMC medical records from January to December in 2018. Workers were from different nationalities and age groups. The ethical approval was obtained from the Ministry of Health and Kuwait University with only the primary researchers granted access to anonymized data.

### 2.3. Diabetes Determination

This study is testing diabetes mellitus as the dependent variable. At SIMC, the diagnosis of diabetes is made when the worker meets one of the following criteria for diabetes: either diagnosed by a physician or at least two elevated serum fasting glucose levels in addition to elevated HbA1c despite adhering to a calorie-restricted diet. On the other hand, information on the independent variables, which are BMI, age, hypertension, nationality, smoking status, and work category, were collected from a questionnaire provided to workers who attend SIMC.

Data was entered into the database by SIMC’s nursing staff. In terms of the quality control of the database, data were entered and then verified by the registration staff. In case of any missing data, the hardcopies of the medical files were checked, or a call was made directly to the related worker.

### 2.4. Statistical Analysis

SPSS version 24 (IBM Corp., Armonk, NY, USA) was used to analyze the secondary data set. All variables used in this study were categorical. For the analysis of the categorical variables, which are: diabetes (no/yes), BMI > 33, BMI category, age category, hypertension, nationality, and smoking (no/yes), frequencies and percentages were used. Additionally, chi square test was used to assess the relationship between the outcome (diabetes) and other covariates. Finally, logistic regression was used to assess the relationship between diabetes mellitus and the other independent factors where odds ratios with confidence intervals were delineated. Univariate regression was specifically used to assess the relationship between diabetes and each other covariates at a time while multivariate logistic regression was applied to illustrate the relationship between diabetes and the other tested covariates and to adjust for all possible confounders that might affect the studied outcome. The 5% significant level was used for the *p*-value. All statistical analyses were performed using SPSS version 24. The codes used to perform the analysis in SPSS for the tested variables are shown in Appendix A.

## 3. Results

### 3.1. Demographic Characteristics of the Study Participants

Information on a total of 3477 participants was recorded in the dataset of 2018. The characteristics of the study participants are summarized in Table 1. Of the total participants, 10.1% had diabetes mellitus. The body mass index characterization of the participants was illustrated in two ways:According to the confined space requirement, where the workers’ BMI should not be greater than 33 to be fit for any tasks in confined spaces. The results showed that most participants were lower than 33.According to the general classification of BMI, where 15–19.99 is considered underweight, 20–24.99 is normal, 25–29.99 is overweight, and >30 is obese. The data showed that almost 49% of the participants were overweight.

Age was classified into 10-year categories. The analysis found that 36.5% of the participants were between 31 and 40 years of age, followed by 26.5% between 41 and 50 years of age.

The findings showed that a small percentage of the participants were diagnosed with hypertension at 11.8% of the total participants. Additionally, 76.1% of the total participants reported themselves as non-smokers.

Regarding the nationality distribution of the study participants, the highest number of participants came from India, Egypt, and Bangladesh. As mentioned earlier, even though the study focuses on migrant workers, Kuwaitis were included, which represents 2% of the total participants, only to report their health status at the time of data collection.

### 3.2. Prevalence of Diabetes Mellitus among the Workers from SIMC

Even though almost half of the participants were overweight, the BMI was not statistically significant. As a matter of fact, the highest prevalence of diabetes mellitus was recorded among those with normal BMI (20–24.99) at 11.2%.

To illustrate the relationship between DM and age, the analysis showed that the prevalence of diabetes mellitus significantly increased with age. As it rocketed from 1.3% in the 21–30-year age group to 30.4% in the 61–70-year age group.

Furthermore, hypertension and nationality were found to be significantly associated with diabetes (*p* < 0.001), where the prevalence of DM among those with hypertension was 15.2% higher than those with no hypertension. When looking closely among the different nationalities of the migrants in this study, although Indians were the largest number of the study participants, Bangladeshis recorded the highest prevalence of DM at 16.2%. Pakistanis came in second place with a prevalence of 15.5%, followed by Indians at 10.5%. It is worth mentioning that among the 14 Kuwaitis in this study, the prevalence of DM reached 20%.

On the other hand, there was no statistically significant difference between smoking tobacco and diabetes.

### 3.3. Measure of Association between Tested Variables and Diabetes Mellitus

An association between DM and the different independent variables among workers from SIMC was assumed but after adjusting for confounders, the association between DM and BMI was found to not be significant (Table 2). Similarly, there was no association between DM and smoking tobacco (AOR = 1.066; 95% CI: 0.792–1.434, *p*-value = 0.673).

On the other hand, there was a significant association between DM and age. Older migrant workers seemed to be more likely to have diabetes (AOR (95%CI): ranging from 3.045 (1.478–6.277) for age group 31–40 to 27.453 (11.205–67.260) for age group 61–70, all *p*-values < 0.05). Moreover, hypertension was also significantly associated with DM (AOR = 1.831 (1.381–2.427)).

Looking closely at the association between DM and nationalities, the results vary between each nationality. Egyptian migrant workers were found to be significantly protected from DM and they are less likely to be diabetic (AOR = 0.500 (0.301–0.829)). Workers from Bangladesh were found to be significantly associated with an increased risk of DM, unlike Filipino, Pakistani, and the other nationalities, their AOR (95% CI) were AOR = 1.476 (1.012–2.150), AOR = 0.705 (0.405–1.228), and AOR = 0.707 (0.452–1.105), respectively. It is worth mentioning that Kuwaitis were found to be at higher risk of having diabetes.

## 4. Discussion

This is the first study in Kuwait to assess DM and its associated risk factor among migrant workers. Migrant workers are usually employed in “3D jobs” which refers to dirty, dangerous, and difficult. The majority of migrant workers occupy professions that do not match their professional profile of the immigrant. They tend to work longer hours in risky jobs when compared to local workers [10]. As mentioned by Barrak Alahmad, migrant workers are sometimes neglected, so justice needs to be brought for this vulnerable group [5].

BMI among the migrant workers was found to be independently not associated with DM. The above finding was inconsistent with the literature review. A cross-sectional study conducted to assess the relationship between BMI and DM among veterans with spinal cord injuries and disorders found that participants who had diabetes were significantly older with a higher mean of BMI values. After adjustment for the race, smoking status, and age, BMI was associated with a 4% higher prevalence of DM for every unit increase in the BMI [11]. Moreover, Nguyen et al.’s study concluded that, among their study population, 13.6% were diabetic, of which, 80.3% were considered overweight (BMI > 25) and about 50% of the diabetic participants were considered obese with a BMI of more than 30. In a follow-up study conducted on more than 445,000 participants in the UK showed that the participants with the highest BMI value have an 11-fold increase in the risk of developing DM compared to those with lower BMI values. Moreover, those with the highest BMI also recorded the highest probability of developing DM, regardless of the effect of genetics. One of the distinctive findings of this study was that, when crossing a specific BMI threshold that differs between one individual and another, the elevated BMI will not have a greater impact on developing DM. The likelihood of developing DM among people with high BMI remains the same regardless of the duration or the increase in weight [12].

On the other hand, Han and Boyko discussed the possible explanation for the “obesity paradox”, which is the inverse relationship between mortality and obese diabetic. They found that despite the risk of obesity in developing DM, the mortality is lower among obese diabetics compared to normal weight diabetics. Han and Boyko pointed out that the obesity paradox might be due to limitations in the epidemiological study or in the use of BMI, which might be the case in our study. As a single measure of the BMI cannot show the real reflection of the person’s weight history, the relationship between body fat and BMI can be affected by other covariates, for instance, muscle mass, body fat distribution, sex, ethnicity, and age (Han and Boyko, 2018). Furthermore, in 2020, Nilsson, Korduner, and Magnusson, raised the concept of metabolically healthy obesity where they discussed that even though obesity is associated with many morbidities and mortalities, obesity does show heterogeneity [13]. As in our findings, there was a large number of participants labeled as obese, in contrast, a small percentage of them appear to have diabetes. Nilsson, Korduner, and Magnusson’s study suggested that metabolically healthy obese are not necessarily protected from metabolic disorders, but more likely to be temporarily protected and the risk factors will be seen soon [13]. According to a cross-sectional study conducted in Southern Sweden that aimed at assessing the characteristics of healthy severe obesity, the metabolically healthy obese was defined as people that do not seek hospital care for many years in their mid-life and did not develop diabetes. The study concluded that obese people have an active physical life that makes them fat but at the same time fit [14]. A well-known example of metabolically healthy obesity was illustrated by Denis and Hamilton, where they mentioned Sumo wrestlers who may weigh almost three times a normal average person and consume about seven thousand calories a day but yet do not suffer from any metabolic disorders. This is due to the daily intense physical exercise, that allows them to store fat just underneath the skin rather than visceral fat [15]. The same mechanisms might apply for our participants, in addition to the healthy worker effect (HWE), which is defined as “a phenomenon initially observed in studies of occupational diseases: Workers usually exhibit lower overall death rates than the general population because the severely ill and chronically disabled are ordinarily excluded from employment” [16], that makes them healthier than the general population. The migrant workers in this study have an active lifestyle that protects them from developing diabetes despite their high BMI. To support that, many studies have shown that physical exercise and a healthy diet reduce the risk of diabetes by 50% [17,18].

Our findings on the prevalence of DM that increased dramatically with age is consistent with the literature review. In fact, almost all studies proved the strong association between aging and the increased risk of DM. For instance, a study was conducted to assess the association between age, body mass index, and the prevalence of diabetes and how it can be affected by ethnicity. The result of the study showed that the prevalence of DM increased with age in all ethnic groups, with a noticeable increase at the age of 30–45, depending on ethnicity [19]. Moreover, a prospective cohort study carried out by Owen et al. examined the impact of high BMI in early and middle life on the risk of cardiovascular diseases and type 2 diabetes. The study found that the high level of BMI at a younger age will be associated with diabetes incidence in their older age [20].

As previously mentioned, our results showed a strong association between DM and hypertension which is in line with many previous prospective studies. A meta-analysis study concluded that people with elevated blood pressure were at higher risk of developing diabetes mellitus [21]. In line with the above findings, a study was carried out to assess the effect of hypertension on increasing the impact of obesity on developing DM in Kuwait. The study results showed 27% of the participants were diagnosed with DM, of which 41% were diagnosed with hypertension. It also concluded that obese patients with hypertension have a greater risk of the early onset of DM, in comparison with obese patients with no hypertension [22]. It is noticeable that 11.8% of the migrant workers had hypertension, which again supports the phenomenon of the healthy worker effect. In addition to that, the improved living conditions compared to their native homes may contribute to improving their general health, just like a finding from a cross-sectional study conducted in Malmo, Sweden, where they found that immigrant populations have lower blood pressure compared to native Swedes. This was due to the improvement in living and environmental conditions [23].

On the other hand, other studies have suggested that diabetes mellitus may increase the risk of hypertension. A recent study by Sun et al. investigated the bidirectional causal relationship between hypertension and diabetes mellitus. The study found a positive causal effect of DM on hypertension and failed to prove the opposite [24]. Similarly, a cohort study was carried out to study the association between hypertension and DM. After adjustment, the study showed that diabetes was associated with a greater than 40% increase in the incidence of hypertension [25]. Sterhouwer et al. suggested that DM increases arterial stiffness, by which the risk of cardiovascular diseases increases [26].

One of the distinctive findings in this study is the association between DM and nationalities. The association was statistically significant; however, it differs among each nationality in this study. Even though the HWE phenomena do appear in our findings, DM is considered a major health problem in the country of origin of our study participants. Generally, almost all migrant workers in this study are of Asian ethnicity. According to Wong et al., Asians tend to have lower BMI levels when compared to Hispanics, non-Hispanic white, and black. Despite the low BMI level, Asians tend to have a significant prevalence of diabetes and hypertension [27]. Similar results were found that among Indians who make up most of our study participants, a moderate association was found between obesity and type 2 diabetes. Additionally, the “Asian Indian Phenotype” was introduced, where the adverse effect of obesity can be seen among Indians despite the lower BMI. This means that even with normal BMI levels, Indians can still develop DM, insulin resistance, and metabolic syndrome [28]. Egyptians were the second majority in this study. When Egyptians were investigated in terms of the prevalence of DM, it was found that Egyptians have other different significant factors that make them more vulnerable to have DM. For instance, the Egyptian population suffers from obesity, chronic hepatitis C, an unhealthy diet, and a sedentary lifestyle, which all contribute to the occurrence of DM [29]. However, our findings showed a protective odds ratio when it came to the association of DM and nationality, which is most probably due to the HWE. Bangladeshis on the other hand had a significant association between nationality and DM. According to a recent study conducted in Bangladesh to assess the prevalence of DM and its associated risk factors, the prevalence of DM was high among those who were overweight, had hypertension, and increased by age. Not only that, but the study also showed that those of low socioeconomic class are left with undiagnosed DM [30]. Unlike the literature, the results of both workers from the Philippines and Pakistan showed no association between DM and their nationality. For example, a study by Araneta et al. compared the prevalence of DM among Filipino women in the Philippines and Filipino women who migrated to Hawaii and San Diego. The results showed that those in the Philippines had lower BMI levels, DM, and hypertension compared to those in Hawaii and San Diego. As suggested by Araneta et al., the adopted Western lifestyle was behind the increase of DM among migrated Filipinos [31]. Another study was conducted in Pakistan to measure the prevalence of DM of more than 3000 participants, 16.98% of them had diabetes and 10.91% of them were prediabetes [32]. These studies indicate that people from the Philippines and Pakistan are also vulnerable to diabetes.

Unlike the literature review and the large number of the studies reviewed, the association between smoking tobacco and DM in this study was found not to be statistically significant. According to a systemic review and meta-analysis, which specifically investigated the association between smoking status, intensity, and smoking cessation on the risk of DM, the result showed that active smokers had a higher risk of developing DM with a pooled relative risk of 1.38 (95% confidence interval (CI), that is, (1.28–1.49) when compared to non-smokers. The study also showed a linear increase in the risk of developing DM with the increase in cigarette consumption. When looking at the risk of developing DM after quitting smoking, the results showed that due to increasing weight after quitting smoking the risk of developing DM considerably increases by about >15% [33]. Additionally, a recent study to assess the association of smoking initiation and DM found a causal association with an odds ratio of 1.28 (95% confidence interval, 1.20, 1.37; *p* = 2.35 × 10^−12^) [34]. Zhang et al. suggested that not only active smokers are at risk of developing DM, but also passive smokers. According to his prospective cohort study, the risk of diabetes among nonsmokers who were exposed to passive smokers was high, where the relative risk of those who were exposed occasionally and regularly was 1.10 (95% CI, 0.94–1.23) and 1.16 (1.00–1.35), respectively [35].

## 5. Limitations

This study has several limitations. The cause-and-effect relationship cannot be determined through cross-sectional study, nor a good representation given of the general health status of migrant workers. Adding to that, the dependence on cross-sectional studies may lead to biases that include the HWE. Moreover, the use of a single measure of BMI as a measure of adiposity is not ideal and needs to be replaced or be taken on a longer scale. For future studies, it is recommended that the level of physical activity needs to be recorded because of physical activity on our outcome. In addition, the economic burden of diabetes and its complication on migrant workers needs to be investigated.

## 6. Conclusions

This is the first study in Kuwait to assess DM and its associated risk factor among migrants since migrant workers are neglected subpopulations that need our focus and attention to achieve justice and fairness. The findings revealed that the prevalence of DM among our study population was considerably lower, which strongly supports the phenomenon of HWE. Migrant workers obviously do not have the same level of physical exercise as Sumo wrestlers. However, the lesson here is that to prevent diabetes, you do not have to engage in high-intensity exercise. Thus, the key is to exercise on a regular basis. Most of our study participants were overweight. Moreover, despite the negative association between BMI and DM in this study, losing weight is highly recommended to maintain health in the long term and avoid any permanent damage in the body caused by obesity. Age and hypertension, on the other hand, were associated with DM in this study, in addition to ethnicity. Thus, maintaining a healthy lifestyle and being physically active will reduce the risk of developing DM. Smoking tobacco, as a risk factor, was not associated with DM among our study participants. Nevertheless, smoking cessation is a must for the prevention of any health complication, including DM. Lastly, DM is a preventable disease. With the increased incidence of DM worldwide and rising percentage of obesity, governmental actions need to be undertaken to reduce its global burden.

## Figures and Tables

**Table 1 ijerph-19-03943-t001:** Diabetes mellitus and demographic characteristics of the study participants from SIMC dataset, Kuwait, 2018.

Demographic Characteristics	Frequency	
	n	(%)
	3477	(100)
DM		
Yes	350	(10.1)
BMI according to confined space		
<33	3195	(91.9)
>33	280	(8.1)
BMI (kg/m^2^)		
15–19.99	77	(2.2)
20–24.99	948	(27.3)
25–29.99	1709	(49.2)
>30	740	(21.3)
Age (years)		
up to 20	0	(0)
21–30	712	(20.5)
31–40	1270	(36.5)
41–50	921	(26.5)
51–60	517	(14.9)
61–70	56	(1.6)
>71	1	(0)
Hypertension		
Yes	412	(11.8)
Nationality		
Indian	1976	(56.8)
Egyptian	483	(13.9)
Bangladesh	271	(7.8)
Filipino	181	(5.2)
Pakistani	142	(4.1)
Kuwaiti	70	(2.0)
Other Nationalities	354	(10.2)
Smoking		
Yes	831	(23.9)

**Table 2 ijerph-19-03943-t002:** Crude and adjusted odds ratios of DM for selected independent variables among workers from SIMC, 2018, Kuwait.

Demographic Characteristics	Sub-Group	Crude Odds Ratio of DM			Adjusted Odds Ratio of DM		
	n	OR	(95% CI)	*p*-Value	AOR	(95% CI)	*p*-Value
	3477						
BMI according to confined Space							
<33	3195	1.0	(Reference)		1.0	(Reference)	
>33	280	0.907	(0.596, 1.380)	0.649	1.014	(0.591, 1.741)	0.958
BMI category (kg/m^2^)							
15–19.99	77	1.0	(Reference)		1.0	(Reference)	
20–24.99	948	9.568	(1.317, 69.51)	0.026 *	6.684	(0.898, 49.758)	0.064
25–29.99	1709	8.560	(1.183, 57.978)	0.033 *	5.209	(0.703, 38.607)	0.106
>30	740	7.940	(1.087, 57.978)	0.041 *	4.421	(0.584, 33.483)	0.150
Age category (years)							
21–30	712	1.0	(Reference)		1.0	(Reference)	
31–40	1270	3.068	(1.496, 6.291)	<0.001 *	3.045	(1.478, 6.277)	0.003 *
41–50	921	15.439	(7.821, 30.479)	0.002 *	13.571	(6.817, 27.017)	<0.001 *
51–60	517	24.646	(12.389, 49.028)	<0.001 *	20.271	(10.090, 40.727)	<0.001 *
61–70	56	34.048	(14.266, 81.264)	<0.001 *	27.453	(11.205, 67.260)	<0.001 *
>71	1	0.000	(0.000)	1.000	0.000	(0.000)	1.000
Hypertension							
No	3065	1.0	(Reference)		1.0	(Reference)	
Yes	412	0.090	(2.635, 4.445)	<0.001 *	1.831	(1.381, 2.427)	<0.001 *
Nationality							
Indian	1976	1.0	(Reference)		1.0	(Reference)	
Egyptian	483	0.367	(0.229, 0.588)	<0.001 *	0.500	(0.301, 0.829)	0.007 *
Bangladesh	271	1.648	(1.157, 2.346)	0.006 *	1.476	(1.012, 2.150)	0.043 *
Filipino	181	0.824	(0.484, 1.404)	0.477	0.705	(0.405, 1.228)	0.217
Pakistani	142	1.558	(0.967, 2.510)	0.068	1.282	(0.768, 2.139)	0.343
Kuwaiti	70	2.125	(1.163, 3.884)	0.014 *	2.327	(1.167, 4.640)	0.016 *
Other nationalities	354	0.674	(0.441, 1.030)	0.068	0.707	(0.452, 1.105)	0.128
Smoking							
No	2646	1.0	(Reference)		1.0	(Reference)	
Yes	831	0.824	(0.628, 1.080)	0.160	1.066	(0.792, 1.434)	0.673

* Significant at the 5% level. The categorical association is conducted using the global chi-square test.

## Data Availability

Restrictions apply to the availability of these data. Data was obtained from Shuaiba Industrial Medical Center (SIMC) and are available (from the corresponding author) with the permission of (SIMC).

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
