# Peer review of "Diabetes Mellitus and Its Risk Factors among Migrant Workers in Kuwait"

_ijerph, 2022, doi:10.3390/ijerph19073943_

Round 1

Reviewer 1 Report

Manuscript ID: ijerph-1528209

Manuscript Title: Diabetes Mellitus and its Risk Factors among Migrant Workers in Kuwait

Authors: Anwar Ali et al.

According to the authors, the present manuscript aims to estimate the prevalence of diabetes and associated risk factors to draw recommendations and suggestions for migrant workers in Kuwait. It is an interesting approach; however, many associated risk factors are not examined. Moreover, many of the suggestions and conclusions of the study are not supported by the current study design and findings, limiting the novelty, generalizability, and impact of the study.

The following points should be considered.

Major Comments 

  1. Please clarify the criteria used for diabetes diagnosis. Moreover, why did the authors require three abnormal tests "at least two elevated serum fasting glucose levels in addition to elevated HbA1c" for diabetes diagnosis? What do they mean with the phrase "despite adhering to a calorie-restricted diet"? Please also add the relevant reference(s). Did the authors evaluate the presence of symptoms of hyperglycemia?
  2. Why did the authors choose the "BMI>33" as a cut-off point? Did the authors consider using BMI as a continuous variable and conduct a sensitivity analysis?
  3. Why did the authors use 19.99 as a threshold instead of 18.5 (standard WHO BMI classification categories) for the BMI subcategories? In addition, why did the authors use the BMI category:15-19.99 as a reference category instead of the BMI category: 20-2,4.99, which falls within the healthy weight range?  
  4. Could the authors define the diagnostic criteria used for hypertension and add the relevant reference?  
  5. Could the authors summarize the inclusion and exclusion criteria of their study?
  6. Based on Table 1 and Appendix A, why did the authors not exclude participants with Kuwaiti nationality since the present study focuses on migrant workers? This should be an exclusion criterion. Otherwise, it would be helpful to compare migrant workers with Kuwaiti workers to explore any potential differences.
  7. It would be helpful if the authors could include the following parameters in their analyses (and in the Tables): age, lipid profile, family history of diabetes, sex, and dietary habits.
  8. It would be useful if the authors could present demographic characteristics (Table 1) between study participants with or without diabetes and present any significant differences between the two groups.
  9. According to the authors' conclusions: "since migrant workers are neglected subpopulation that need our focus and attention to achieve justice and fairness". However, the authors did not explore possible disparities in accessing health facilities, social services, education, affordable healthy food, or other environmental disadvantages.
  10. Moreover, the authors mention: "The findings revealed that the prevalence of DM among our study population was considerably lower, which strongly supports the phenomenon of HWE". Would it be possible for the authors to clarify how the present study's findings and design strongly support this statement?
  11. According to the authors: "Migrant workers don't have obviously the same level of physical exercise as Summo wrestlers. But the lesson here is that in order to prevent diabetes, you don't have to engage in high intensity exercise", and "Sumo wrestlers who may weight almost three times a normal… The same mechanisms might apply for our participants," (lines 227-231). How do the authors support this conclusion based on the current study design and analysis? Moreover, did the authors explore the level and intensity of the work to draw such a conclusion?
  12. According to the authors: "Moreover, despite the negative association between BMI and DM", do the authors mean the absence of association?

Reviewer 2 Report

Bibliographic references must be adapted to the journal's standards (Vancouver style).

Author Response

Bibliographic references must be adapted to the journal's standards (Vancouver style).

Agree

Reviewer 3 Report

This study investigate the prevalence of diabetes mellitus in a specific population of migrants in Kuwait. This study is interesting but has many methodological weaknesses. Here are some thoughts to improve your paper.

Some points on the form of the article needs to be revised. The authors should be more consistent in using the term diabetes mellitus. If they choose to use an abbreviation, here DM, please use it throughout the article. Furthermore, authors should revise their texts for typos, references and for the wording of certain sentences.

Few questions about the study design: was a research protocol developed for this study? if so, is it available? Did the authors use the appropriate guidelines to improve the reporting of their article (i.e. the STROBE guideline, https://www.equator-network.org/reporting-guidelines/strobe/)? Did the authors calculate a sample size for this study to ensure sufficient statistical power to perform the analyses?

Regarding the methodology used for this study, too little information is provided on the characteristics of the patients and how they were selected. What were the inclusion criteria? Age, sex, reasons for consultations, workers in a specific domains (and which one)? How did you get their consent? Did they all agree to participate in epidemiological studies during their prior consultations? The authors should provide more information about the participants and the selection process.

Then, in the data collection, the authors refer to ‘secondary data’. Why secondary? What was the primary data? In this paragraph, the authors should also describe in detail the data set collected and why these data are important to investigate the link with DM. For example, the authors should describe why they used two classifications of BMI or explain what are the ‘confined space requirement’ and why is it important in this study.

In the sociodemographic characteristics of the population, why is the gender of the participants not presented? Couldn't gender be a potential risk factor? The authors should add this crucial information in their analyses. Why is there no information on the different types of work done by migrants? This could be very interesting to get an idea of the energy expenditure during their work.

Concerning the statistical analyses, I do not understand all the authors' choices on the variables chosen and the reference categories used. For example : why use the undernutrition category as a reference? The category of normal weight seems to me more appropriate. Same for nationality, since the Kuwaiti cannot really be considered as migrants, why not use this category as a reference? Why did you choose the Indian?

Still in the results, I'm not sure about the relevance of table 2. It does not provide any more information than table 1 and 3. It would be interesting to find a way to summarize them.

In the discussion, the authors stated “BMI among the migrant workers was found to be independently not associated with DM. The above finding was inconsistent with the literature review”. One explanation may be the choice of the reference categories. See my comment below. Throughout the discussion, the authors refer to the "HWE" phenomenon. On what scientific basis could they determine that their population could be classified as such? More information is needed!

Round 2

Reviewer 1 Report

Manuscript ID: ijerph-1528209 (Revised version)

Authors: Anwar Ali et al.

Manuscript Title: Diabetes Mellitus and its Risk Factors among Migrant Workers in Kuwait

The authors have tried to respond and address the comments. No further comments.